# Using Gaze for Behavioural Biometrics

**DOI:** 10.3390/s23031262

**Published:** 2023-01-22

**Authors:** Alessandro D’Amelio, Sabrina Patania, Sathya Bursic, Vittorio Cuculo, Giuseppe Boccignone

**Affiliations:** 1PHuSe Lab, Department of Computer Science, University of Milano Statale, Via Celoria 18, 20133 Milan, Italy; 2Department of Psychology, University of Milano-Bicocca, Piazza dell’Ateneo Nuovo 1, 20126 Milan, Italy

**Keywords:** biometric recognition, behaviour characteristics, gaze identification, foraging theory, visual attention, eye movements, stochastic processes, Bayesian inference, machine learning

## Abstract

A principled approach to the analysis of eye movements for behavioural biometrics is laid down. The approach grounds in foraging theory, which provides a sound basis to capture the uniqueness of individual eye movement behaviour. We propose a composite Ornstein-Uhlenbeck process for quantifying the exploration/exploitation signature characterising the foraging eye behaviour. The relevant parameters of the composite model, inferred from eye-tracking data via Bayesian analysis, are shown to yield a suitable feature set for biometric identification; the latter is eventually accomplished via a classical classification technique. A proof of concept of the method is provided by measuring its identification performance on a publicly available dataset. Data and code for reproducing the analyses are made available. Overall, we argue that the approach offers a fresh view on either the analyses of eye-tracking data and prospective applications in this field.

## 1. Introduction

The problem we are confronting in this article, eye movement-based biometrics, is best introduced by considering Figure 1. The figure presents an excerpt from the data to illustrate some characteristics of gaze dynamics that motivate our analyses and the proposed model.

The coloured traces unfolding over the image depict the wandering gaze of three different observers, recorded through an eye-tracking device, while they were scrutinising the imaged scene in the free viewing condition for a 2 second presentation. The wandering pattern of gaze trajectories puts on view that where observers choose to look next at any given moment in time is not completely deterministic, but neither is it completely random [1]. Indeed, under such circumstances, observers’ attention is by and large deployed to semantically relevant parts of the scene [2], e.g., the person’s face; yet, each beholder betrays a personal, somehow idiosyncratic viewing behaviour (Interestingly enough, the term “wandering” has been used in recent research (e.g., [3,4,5]) to characterise eye movements as a marker of mind wandering (self-generated thoughts that are irrelevant to the current task). The prevalence of episodes in daily life of disengagement of attention from external information to internally directed cognition is likely to have practical implications for eye movements biometrics, given its tight connection to the individual’s self and personality [6]).

The rationale behind behavioural biometrics, in general, and particularly the recourse to eye movement analysis as a viable route to a subject’s identification, conceals deep roots beyond its deceptively obvious and, to a great extent, instrumental practice (overviewed in Section 2). All in all, each individual is somehow a bit different than the other one (a difference that might be formalised in terms of physical complexity [7,8,9,10]). Indeed, the individual’s self ontogenetically changes as the result of his/her intimate experience; it sediments going through an active assimilation-accommodation process in which the world (and the social environment) is internally organized in a model that reflects current knowledge about it and the individual’s role in it. The ensuing actions are part of this process and hence altogether incomparably personal, unique, and infinitely individual [10,11,12,13]: the seamless dance of gaze makes no exception [9,14].

**Figure 1 sensors-23-01262-f001:**
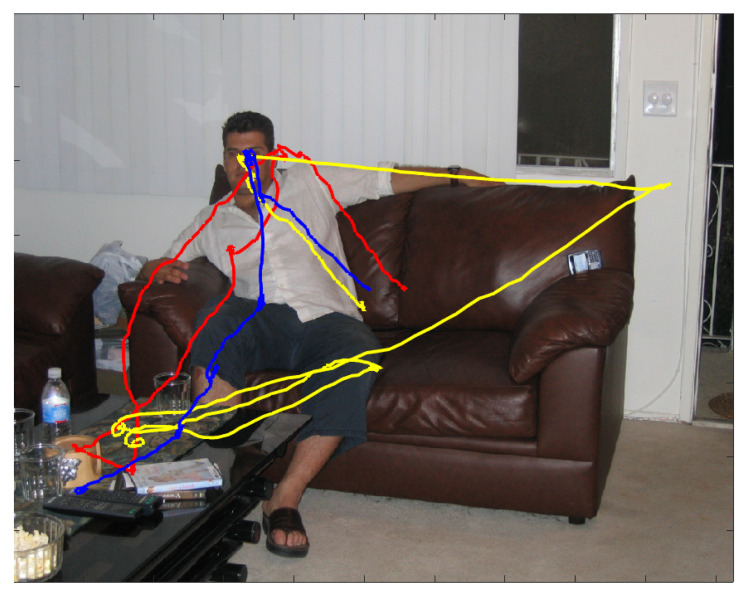
Gaze dynamics of three different observers while scrutinising for 2 secs the image and recorded through an eye-tracking device. The unfolding of gaze deployment of each observer is displayed as a coloured trajectory (blue, yellow and red, respectively) overlapping the viewed image. The onset of gaze unfolding is approximately located at the center of the image for all viewers. Image and eye-tracking data are publicly available from the FIFA dataset [15].

An intimate alliance is thus established between the uniqueness of the self and the gaze, inasmuch as it might be surmised that the way to capture the ineffable something which differentiates qualitatively each of us, as Marcel Proust awesomely phrased it, is to “*possess other eyes, to see the universe through the eyes of another, of a hundred others, to see the hundred universes that each of them sees, that each of them is*” [16].

In this paper, we argue that by unearthing these roots, a sound foundation is offered to eye movement-based biometric methods blossoming under the behavioural umbrella, which paves the way to novel, explainable and principled methods and techniques.

In a crude summary, the chief concern of the work presented here boils down to the following research objectives:To lay down a principled approach to exploit eye movements in behavioural biometrics;To derive, in the context of such a framework, a computational model suitable to operationalize biometric assessment.

The first goal is spelled out in terms of foraging theory as a sound approach to capture the uniqueness of individual eye movement behaviour (see Section 3).

The second is shaped in the compact form of a composite (local and global) Ornstein-Uhlenbeck (O-U) process, whose relevant parameters provide a suitable feature set for biometric identification. The O-U parameters are inferred from eye-tracking data via Bayesian analysis (Section 4 and Section 5). This sets the fundamental research question of this article: is such a succinct phenomenological model of gaze behaviour adequate for biometrics?

A proof of concept is thus put forward in Section 6, by using a publicly available eye-tracking dataset [15]. A typical example of the kind of data we are dealing with has been presented in Figure 1. Data and code for reproducing the analyses reported in this Section are made publicly available.

Eventually, a discussion of results so far achieved is given in Section 7.

To start this journey, the forthcoming section briefly overviews the state of the art of current techniques.

## 2. Related Works

Biometrics deals with the exploitation of unique, identifiable and quantitatively measurable characteristics of humans in order to authenticate and/or identify them.

In the last decade, biometrics has been spreading in our daily life, just consider finger and face recognition in common electronic devices (smartphones, tablets, laptops etc.). Traditionally, biometrical methods have focused on physical characteristics, namely personal physical traits such as fingerprints [17,18], face [19,20], iris and retina [21].

More recently, behavioural traits have been considered as well. Behavioural biometrics explores what kind of behaviour might be unique to a user, resulting in methods generally less intrusive than physical biometrics and well-suited for continuous authentication. Among those behavioural-based approaches, eye movement biometrics (EMB) has proven to have great potential in a variety of contexts. In stimulus-dependent scenarios we can include reading [22,23], static point gazing [24] or jumping point tracking [25,26,27], virtual reality tasks [28,29,30], free viewing of static images [31,32], including faces [33,34]. The more challenging task-independent scenario has been explicitly investigated by Kinnunen et al. [35], whereby no assumptions that the same stimulus appears in training and testing are made. In this direction, other works have exploited the potential of virtual reality (VR) for the emerging use of reliable eye-tracking sensors [29,30].

The first attempt to introduce eye movements in biometrics was undertaken by the pioneering work of Kasprowski and Ober [36]. Support vector machine (SVM), k-nearest neighbors (KNN), decision trees and Bayes classifiers were applied to selected cepstral coefficients of eye movement signal with promising results supporting the endeavor. Still, the analysis was limited to raw trajectory signals, without an explicit classification of eye movement signals into physiologically-grounded events. In [37] eye movements were considered from a higher-level perspective. Fixations and saccades-related features were combined and fed to a probabilistic neural network to extract match scores. In the same vein, Holland and Komogortsev [22] extract a broader set of high-level features then compared using a Gaussian kernel. Moreover, a graph-based model has been proposed with good results [34]. Grounded in statistical analysis as well, the STAR approach by Friedman et al. [38] is considered the state of the art within the framework of hand-crafted features. Following event classification, the method draws on principal component analysis (PCA) and intraclass correlation coefficient (ICC), an index of the temporal persistence and stability of a single biometric feature.

Yet, latterly, the best results have been achieved by deep learning methods, using convolutional neural networks [39,40,41,42], residual networks and densely connected convolutional networks [30]. Jia et al. [39] utilised long short-term memory cells (LSTM) to build a recurrent neural network (RNN) that relies on raw data. Instead, the method proposed by Jäger et al. [40] requires an initial transformation of raw eye movement signals to isolate desired micro eye movements according to their characteristic velocities, with the resulting scaled values fed into a CNN with two separate subnets. Other works, after an initial feature extraction step, explored metric learning for eye movement biometrics [30,42]. The current state-of-the-art model is DeepEyedentificationLive (DEL) [41] which discerns “fast” (e.g., saccadic) and “slow” (e.g., fixational) eye movements then fed into two separate convolutional subnets. Another recent model, Eye Know You (EKY) [30], uses exponentially dilated convolutions to achieve reasonable biometric authentication performance with a relatively small (∼475 k learnable parameters) network architecture.

By and large, the above methods treat eye tracking data by crudely relying on classical analyses (e.g., dwelling time, distance and direction changes between fixation points, spatial distribution of saccades [43]) of the scan paths—i.e., sequences of fixations and saccades, as parsed by some algorithm from the original gaze trajectories. This limitation holds, independently of the subsequent classification techniques (either classic or deep learning-based) adopted. Further, the richness of processes giving rise to gaze behaviour is scarcely acknowledged. As a matter of fact, this approach overlooks a great deal of information that can be useful for determining the uniqueness of behaviour hiding behind the individual’s gaze dynamics.

## 3. Background and Motivation

Individual-based approaches in ecology seek a mechanistic understanding of how variation among individual organisms generates or contributes to patterns at population, community and ecosystem levels (but for an in-depth discussion, see [44]). What makes an organism unique? Why does this animal have this unique set of traits? Clearly, answering such questions involves describing highly specific, idiosyncratic properties and causal histories.

Individuality in behavioural ecology is defined as the phenotypic and ecological uniqueness of single individuals; namely, what makes an individual phenotypically unique is generally not a single phenotypic property, but rather the possession of a whole set of phenotypic traits together with unique sets of ecological relations [44]. Features like behaviour, habitat use, feeding preference, social relations, and so on, are core to what makes individuals unique.

In this broad perspective, one remarkable example is foraging behaviour [45,46]. Foraging is a term that includes where animals search for food and which sorts of food they eat. According to optimal foraging theories (OFT), the forager continuously faces four problems [47]:patch-choice: the forager has to make a decision concerning what type of patch (a clump of food) to search for (e.g., good bushes full of berries);patch-finding: the forager wanders through a landscape where certain resources are available and makes a choice on how to move between patches (optimal movements);target/prey-finding: once a patch is located, the forager should decide what prey to take and handle (optimal diet choice);patch-leaving: the available resources in the patch decrease over time spent handling preys; eventually, a forager makes the decision of leaving the current patch to search in a more profitable patch, or just to finish the task (giving up or departure times from patches).

All in all, the forager, in order to solve the above problems, engages in a recurrent cycle of successive local exploitation of the patch alternated with exploration, i.e., relocation between patches. While engaged in the foraging cycle, the forager also learns about the profitability of the individual patches and the environment after visiting each patch. Such exploitation/exploration pattern is the genuine fingerprint of animal’s foraging behaviour.

In recent research, consistent individual differences in behaviour (animal personality) and food resource use (individual specialization) have emerged. Animal personality and individual specialization are patterns of individual-level phenotypic variation, both describing similar concepts of temporally consistent individuality in behavioural and food web ecology [48]. Personality is a motif which accounts for consistent behavioural differences or traits in animals (tipically: exploration, activity level, sociability, boldness and aggressiveness); indeed, research in behavioural ecology provides evidence that personality traits can be detected in a vast collection of invertebrate and vertebrate taxa [48]. Individual specialization involves significant individual differences in the animal’s diet arising within a population. Most important here, differences in both personality and specialization remain consistent over time and across ecological contexts; namely a steady variation is observed among individuals, as opposed to a strikingly low variability within the individual [48].

Given that foraging behaviour provides an important marker of animal individuality, how does then foraging link to cognition and, in turn, to eye movement behaviour?

In a nutshell, it has been argued that ancestors’ foraging for material resources in a physical landscape evolved over time in foraging for information in cognitive space [49,50,51,52,53]; when the individual specifically engages in a cognitive visual task, then gaze foraging over a visual scene is by far the fundamental action to sample and gauge the visual input [14,54,55,56,57].

At the psychological level, goal-directed foraging has been related to relevant cognitive skills [49]: spatial memory (the ability to recall the location of resources and navigate efficiently between them), value-based decision-making (to choose the best course of action given all the available alternatives, what is), and executive control (the versatile guidance of behaviour, overriding reflexive, automatic responses). Even self-awareness, deliberation, and free will can be subsumed under this view.

At the brain level, the neuro-molecular architectures and mechanisms that support the foraging mind are nothing but the evolutionary response to the exploration/exploitation dilemma that must be faced in the external, physical environment [49]. For instance, it has been surmised by Hill [53] that in both animal and cognitive foraging, dopamine acts as a modulator between two behavioural extremes: when dopaminergic activity is high, behaviour is focused, and eventually, stereotypic; by contrast, when dopaminergic activity is low, behaviour becomes unfocused and fails to endure  [53]. Noticeably, Hill’s dopamine hypothesis fits comfortably in recent perspectives on visual foraging based on the active inference account [14,58], an influential framework in current theoretical neurobiology. Here, in a Bayes-optimal behavioural view, dopamine controls the precision or salience of (external or internal) cues that engender actions, either exploitative (i.e., risk minimising) and explorative (i.e., ambiguity minimising) [14].

To sum up, foraging as exploration/exploitation trade-off is an appealing and principled framework for dealing with gaze at different explanation levels  [14,55,56,57].

This statement can be intuitively appreciated at a glance by considering the single trace in the left panel of Figure 2: gaze dynamics unfolds by alternating explorative, long relocations between semantically relevant image patches (e.g., the face), and local movements to “exploit” the selected patch. Importantly, these trajectories represent the microscopic, fine resolution scale of gaze dynamics. At a coarser scale, this explore/exploit pattern can be parsed into a “saccade and fixate” discrete sequence [59] or scan path (right panel of Figure 2), which is most often chosen as the input for further analyses in either the visual attention/eye movement and the behavioural biometrics fields. Saccades are thus functionally defined as the fast movements that redirect gaze to a new part of the surroundings; each fixation summarises fixational movements that occur within intervals between saccades, in which gaze is held almost stationary, so as to keep onto the fovea (the central part of the retina) the circumscribed region of interest (RoI) within the viewed scene.

To make the connection clear beyond the intuition, Table 1 lays down the relationship between deployment of attention through gaze over a scene and animal foraging behaviour.

It is important to note that the given task and the stimulus exteroceptive properties are not the only constraints to the visual exploration/exploitation pattern exhibited by the perceiver. Interoception—the observer’s own perception of his/her internal state of the body, matters too, and, consequently, the perceiver’s affective state and feelings: it is no surprise that in everyday life, gaze is of service to cogently capitalize on visual information that includes social information; markedly, others’ emotions and intentions [60,61,62].

All in all, the exploration/exploitation pattern springing from the gaze sampling endeavour provides a signature of the individual’s plans, goals, interests, likely sources of rewards and expectations about future events, [63,64], social traits and personality [61,65].

Further, in addition to the exteroceptive/interoceptive properties of the stimulus coupled with the pursued goal/task, one should consider biomechanical factors and motor uncertainty (affecting, for instance, oculomotor flight time and landing accuracy) as relevant sources of systematic tendencies or “biases” [66]. The latter, albeit often neglected in the literature, can be conceived as those regularities that are common across all instances of, and manipulations to, behavioural tasks [66,67]. One remarkable example is the amplitude distribution of saccades and microsaccades (ocular movements occuring within a fixation) that typically displays a positively skewed, long-tailed shape [66,67,68,69].

Such a multifactorial drive gives rise to a tiny probability that two subjects will fixate exactly the same location of the visual scene at precisely the same time. Such variability is easily detected in a variety of experimental conditions, especially under a free-viewing or a general-purpose task: e.g., when looking at natural images, movies [69], or even dynamic virtual reality scenes [70]. Interestingly enough, this effect is more pronounced when free-viewing static images (but see [68], for a discussion), as it has been shown in Figure 1: consistency in fixation locations selected by observers decreases over the course of the first few fixations after stimulus onset. Crucially, variations in individual scan paths (as regards chosen fixations, spatial scanning order, and fixation duration) still keep on when the scene embeds semantically rich “objects" and get to be idiosyncratic [68].

On the other hand, recent studies considered the variability of eye movements between observers marking off the characteristics that are stable and reliable; these should be conveniently handled as a trait of the observer rather than be downgraded to “noise” [71,72]. For example, Guy et al. [61] have shown that the amount of time subjects fixate on others’ faces differs between individuals in a consistent manner; this has been identified as a novel gaze-related trait

In these terms, gaze behaviour is apt to unveil individual characteristics tantamount to gait and speech. Different from gait and speech, however, gaze behaviour ineluctably controls the visual input processed by the brain, hence making such variability an important factor in determining and reflecting the individual’s inner world [9,14]. Given such circumstances, the forthcoming Section is devoted to spelling gaze dynamics in mathematical terms so that the gauging of its individual peculiarities can be eventually operationalised.

## 4. Modelling Eye Movements Dynamics

Consider the gaze trajectory shown in Figure 3. Each gaze position at time *t* can be represented by the time-varying location vector x(t) (screen coordinates). Thus, each observed trajectory {x(t),t=0⋯T} can be intended as a realization of a stochastic process {X(t),t=0⋯T}, with X(t)=x(t), namely a random walk (for simplicity, in the sequel, with some abuse of notation, we will use x to denote both the random variable (RV) X(t) and its realization).

The dynamical law governing the evolution in continuous time of such process can be described at the most general level through the *Langevin* Stochastic Differential Equation (SDE):(1)dx(t)=A(x,t)dt+D(x,t)dL(t)

Notably, Equation (Equation 1) is composed by a deterministic component A(x,t), usually called *drift*, and a stochastic one D(x,t), called *diffusion*. Here $L\left(t\right)=\int_{0}^{t}\eta \left(t^{\prime }\right)dt^{\prime }$ is the noise process driven by a suitable probability density function η∼P(η).

The idea of modelling eye movements as stochastic processes is not new; among others, two interesting examples are the modelling of fixational eye movements by [73,74,75,76] and the study of saccades amplitude by Brockmann and Geisel [77]. Engbert and colleagues [73,74,75,76] analyzed the random walk behaviour of fixational eye movements (FEMs) by describing it via the mathematical formalism of fractional Brownian Motion.

Brockmann and Geisel [77] proposed a phenomenological model where gaze shifts are accounted for by Lévy Flights (LF); these are a class of stochastic processes whose innovation term is represented by a distribution with infinite variance (typically an α-Stable distribution).

The microscopic description of a pure LF is straightforward; these are Markov processes with stochastic increments sampled from an α-Stable distribution. The SDE describing a LF can thus be obtained from Equation (Equation 1) by setting A(x)=0, and setting dL(t)=ψ(t)dt with ψ(t)∼f(ψ;α,β,γ,δ), where f(ψ;α,β,γ,δ) represents an α-Stable distribution:(2)x(t)=D(x,t)dL(t)

The different parameters are, respectively, the skewness $\beta \in \left[-1;1\right]$ (a measure of asymmetry), the scale γ>0 (width of the distribution), the location δ∈R and, most important, the index of the distribution (or characteristic exponent) α, that specifies the asymptotic behaviour of the distribution [78,79]. Notably, by setting $\alpha =2,\gamma =\frac{1}{\sqrt{2}},\delta =0$ and $b\left(x,t\right)=\sqrt{2D}$, standard Brownian Motion is recovered.

The approach was operationalised by Boccignone and Ferraro [80]. Actual gaze behaviour was modelled as a Constrained Lévy Exploration (CLE), where the drift component of Equation (Equation 1) is written in terms of the gradient of a potential from the saliency map and the noise component is sampled from a Cauchy distribution (i.e., α=1). Albeit the simplicity of the model, a recent review by Kummerer et al. [81] shows that CLE outperforms a number of more complex models, given an appropriate saliency map. Strikingly, CLE still exhibits a remarkable performance even in the case a constant salience is provided as input [82].

### 4.1. Observers as Foraging Animals

Crucially, such view allows to connect eye guidance modelling to the theories of animal foraging in the ecology literature, by means of what has been referred to as the *Lévy Flight Foraging Hypothesis* (LFFH) [83]. According to LFFH since Lévy Flights maximize the amount of space covered in a fixed time period, they optimize random searches. Hence, natural selection should have led to adaptations for LF foraging. This is due to their super-diffusive nature arising from the power-law decay of the tails of the step amplitude distribution. Stated differently, as anticipated in Table 1, one could think of the eye (or the brain networks controlling eye behaviour) as a forager searching for valuable information (preys) in a given (and possibly time varying) scene (foraging landscape). The LFFH has been applied as modelling choice in different studies concerning eye movements, for simulation or descriptive purposes (see [84], for an in-depth discussion).

Despite being compelling and grounded on first principles, the LFFH approach presents at least two issues:Coarse-grained representation of saccadic movements: saccades are assumed to be ballistic motions, i.e., the path from two successive fixations points is assumed to be a straight line. This assumption is clearly oversimplified (e.g., see again Figure 3).Mathematical hindrances: LFs rely on the use of α-stable distributions as the innovation term; such representation involves dealing with infinite variance and the absence of a closed-form probability density function (PDF).

More on the methodological side, in spite of the large success of the LFFH in the ecological literature, the general applicability of such theory is still the subject of some controversy [85,86]. In fact, many of the observed patterns that are attributed to Lévy processes can be generated by a simpler composite random walk process where the turning behaviour is spatially dependent [85,87,88].

In particular, [85] argued about the fact that even if the observed patterns of movement might resemble those of LFs (step length frequency distribution well fitted by a straight line in a log-log plot), the real generating process need not be a LF process. The author shows how the Composite Brownian Walks (CBW) process generates search patterns that mimic those generated by LF processes; moreover, under specific circumstances the former are more efficient than LFs [85].

CBWs are obtained as a mixture of two “classical” random walks, i.e., stochastic processes characterized by a step length distribution whose variance is finite, which act jointly to mimic the intensive and extensive search of foraging animals. One of such processes is in charge of producing frequent and exponentially distributed steps with relatively small mean, while the other produces more sporadic exponentially distributed steps with relatively large mean.

In the same vein of [85], one could argue about the plausibility of considering eye movement dynamics (the alternating of fixations and saccades) as generated from a mixture of simpler random walks, rather than LFs.

In support of such claim, consider again the scan path depicted in Figure 3. It is worth noticing how the saccades are not ballistic displacements but exhibit some curvature and randomness (Figure 3, right), hence these can be conceived as biased random walks towards the arriving location (the attractor). The same observation holds for fixational movements, which can be seen as stochastic processes with an attractor represented by the center of the fixation location (Figure 3, left). Interestingly enough, it has been argued that a single model of oculomotor behavior, yet enacted at different scales, can explain the continuum from saccadic exploration to fixation [89,90,91].

### 4.2. Observers as Random Walkers—The Ornstein-Uhlenbeck Process

A compact description of both fixational movements and saccadic displacements in terms of a particle randomly wandering but being pulled towards an attractor can be formalised via the Ornstein-Uhlenbeck (O-U) process [92,93]. This can be obtained from Equation (Equation 1) by setting the drift A(x,t)=B(μ−x(t)), the diffusion D(x,t)=Γ and the innovation term η∼N(0,1). Thus,
(3)dx(t)=B(μ−x(t))dt+ΓdW(t).

Here x(t) represents the position of the trajectory in a 2-dimensional space, which is pulled towards the steady state represented by the 2-dimensional vector μ, called the attractor. The latter represents the arriving point of a saccade to the selected fixation location. The adjustment to the attractor is now determined by the 2×2 matrix B providing the magnitude of the attraction effect,
(4)$$B=\left[\begin{array}{cc} B_{ii} & B_{ij}\\ B_{ji} & B_{jj} \end{array}\right].$$
$B_{ii}$ and $B_{jj}$ represent the drift of the process towards the attractor in the *i* (horizontal) and *j* (vertical) dimensions, respectively, while the off-diagonal elements $B_{ij}=B_{ji}=\rho_{B}\sqrt{B_{ii}B_{jj}}$ encode the cross-correlation between drift in both dimensions. For high values of $B_{ii}$ or $B_{jj}$ a faster change will occur in the direction of the attractor. For high values of the cross-correlation, an increasing value of the drift in one dimension will positively affect the drift in the other, producing more curved trajectories towards μ. Based on this properties, the parameter is often called the dampening force or centralising tendency. In order to ensure the stability of the process (convergence to the stationary distribution) the matrix B is supposed to have all positive eigenvalues [94].

On the other hand, the 2×2 covariance matrix Γ controls the variances and covariances of the two driving white noise processes described by dW(t). Higher values of variances/covariances will produce noisier/more anisotropic gaze trajectories.

Notably, such view allows to treat fixations and saccades as realisations of processes exhibiting the same dynamics, namely an O-U process. Following [95,96], the sequence of fixations and saccades composing a scan path can be described via a state-dependent SDE defining a 2D process.

Denote $s_{t}\in \left[fix,sac\right]$ the state variable indicating whether at time *t* either a fixation or a saccade is performed [95,96]; in other words, $s_{t}$ is a switching binary RV accounting for the transition from the current fixation to the next saccade and vice versa. Then, Equation (Equation 3), can be written in the state-dependent form:(5)$$dx\left(t\right)=B^{s_{t}}\left(\mu^{s_{t}}-x\left(t\right)\right)dt+\Gamma^{s_{t}}dW^{s_{t}}\left(t\right),$$
where $B^{s_{t}},\mu^{s_{t}},\Gamma^{s_{t}}$ and noise $dW^{s_{t}}\left(t\right)$ are now the state-dependent parameters and noise process, respectively; in simple terms, depending on $s_{t}$, Equation (Equation 5) can describe either local fixational dynamics or global, saccadic displacements.

Recently, theoretical models have been proposed to simulate gaze behaviour based on composite O-U processes in the framework of either a stochastic version of the Marginal Value Theorem [95] or of multi-alternative perceptual decision making [96]. However, the advantage of having general solutions derived from first principles is mitigated by both the complexity in determining the parameters and the computational cost that are likely to burden biometric applications.

Here, differently, we propose a method which simply relies on a straightforward Bayesian inference of process parameters, which is detailed in the following Section.

## 5. Proposed Approach

In the previous Section we have seen that gaze dynamics $x\left(t\right)\to x\left(t^{\prime }\right)$, with $t^{\prime }>t+\delta t$, δt being an arbitrary time step, unfolds in time according to the law described by Equation (Equation 5). Given the set of parameters $\theta =\{s_{t},B^{s_{t}},\Gamma^{s_{t}},\mu^{s_{t}}\}$, the simulation of a sequence of eye movements can be obtained by solving Equation (Equation 5). In generative form, the solution can be written as the conditional sampling of $x(t^{\prime })$ given x(t), i.e., $x\left(t^{\prime }\right)|x\left(t\right)∼P\left(x\left(t^{\prime }\right)|x\left(t\right)\right)$, where the distribution P(·) is the Normal distribution N(·) (see e.g., [97,98]):(6)$$x\left(t^{\prime }\right)|x\left(t\right)∼N\left(\mu^{s_{t}}+e^{-B^{s_{t}}\delta t}\left(x\left(t\right)-\mu^{s_{t}}\right),\Psi^{s_{t}}\right),$$
where Ψ=$D^{s_{t}}-e^{-B^{s_{t}}\delta t}D^{s_{t}}e^{-B^{s_{t}T}\delta t}$; $B^{s_{t}}$ and $D=\frac{\Gamma^{2}}{2}B^{-1}$ are 2×2 matrices and the form $e^{-M}$ denotes the matrix exponential.

The inference of model parameters θ makes available a concise description of the dynamics of the observer’s gaze. Crucially, the chief research question of the present work is indeed related to the possibility that one such description is predictive of the observer’s identity.

The estimate of the parameters of Equation (Equation 6) can be carried out in two steps:Gaze event classification; namely, parse the raw gaze data of one trajectory, in order to identify the sequence of fixations and saccades in that trajectory (usually defined as a scan path, in the literature).SDE parameter estimation for each identified event along the trajectory/scan path.

### 5.1. Gaze Event Classification

To classify raw gaze data into fixation and saccade events, we do not rely on the event data provided in the dataset. More generally, a functional definition of such events [99] is exploited. We consider a fixation as a period of time during which a static or a moderately displacing part of the visual stimulus (the patch) on the screen is gazed at and that in a human observer would be projected to a relatively constant location on the retina. This corresponds to local dynamics in the exploitation stage. Accordingly, saccades are the gaze shifts for redirecting the line of sight to a new patch of interest, as performed along the exploration stage.

This is operationalised using the Naive Segmented Linear Regression (NSLR), a method for eye-movement signal denoising and segmentation, and a related event classification method based on Hidden Markov Models (NSLR-HMM) [100]. The algorithm is used with default settings; the original implementation is available from the online repository (https://gitlab.com/nslr/nslr-hmm, accessed on 14 January 2023). The NSLR-HMM method, which is suitable to operate on image sequences, classifies fixations, saccades, smooth pursuits and post-saccadic oscillations (PSO). Smooth pursuit is not present in our case (static images) and saccades embed PSOs. This allows clustering the raw gaze trajectories into a sequence of *L* events, each one being either a fixation or a saccade.

### 5.2. Bayesian Estimation of SDE Parameters

Since the time step δt of recorded trajectories depends on the sampling frequency of the eye-tracking device, assume a unitary time step δt=1, without loss of generality [101]. Thus, the observed gaze trajectory is represented by the discrete sequence $\left\{x_{i},i=0,\cdots ,n\right\}$ evolving through steps $x_{i}\to x_{i+1}$. A corse-grained description of such trajectory can be provided in terms of scan path, that is the discrete sequence of events (fixations and saccades). The set of parameters defining an event is thus the basic descriptor we are addressing; formally, such descriptor can be inferred as follows.

The gaze trajectory x(t), as parsed by the NSLR-HMM algorithm [100], is composed by *F* fixational events $e^{fix}=\left[e_{1}^{fix},...,e_{F}^{fix}\right]$ and *S* saccadic events $e^{sac}=\left[e_{1}^{sac},\ldots ,e_{S}^{sac}\right]$. Define $\xi =[e^{fix}|e^{sac}]$.

Consider now the slice $x^{e}=\left[x_{m},\ldots ,x_{q}\right]$ of the sample x(t) , with m⩾0 and q⩽n; $x^{e}$ represents a generic event e∈ξ as classified by the gaze parsing algorithm. Note that the event *e* identified at time *t* takes the role of the switching variable $s_{t}$ in Equations (Equation 6) and (Equation 5).

Thanks to the Markovian and Gaussian properties germane to the O-U process, the likelihood of the slice, given the parameters $\left\{B^{e},\Gamma^{e}\right\}$ is:(7)$$P\left(x^{e}|B^{e},\Gamma^{e}\right)=\prod_{i=1}^{q-m-1}P\left(x_{i+1}^{e}|x_{i}^{e},B^{e},\Gamma^{e}\right).$$

Here, $B^{e},\Gamma^{e}$ represent the event specific set of O-U parameters. Note that here we exclude the $\mu^{e}$ parameter from the inference set, due to the fact that it represents either the arriving point of a saccade or the center of a fixation, hence it does not deliver *per se* any contribution to the description of the gaze event dynamics. The posterior probability of the O-U parameters of the event *e* given the gaze trajectory slice follows from Bayes’ theorem:(8)$$P\left(B^{e},\Gamma^{e}|x^{e}\right)=\frac{P\left(x^{e}|B^{e},\Gamma^{e}\right)P\left(B^{e},\Gamma^{e}\right)}{P(x^{e})}.$$

In order to ensure the matrix $B^{e}$ to have all positive eigenvalues, we force it to be a covariance matrix; therefore, the Lewandowski-Kurowicka-Joe distribution [102,103] is adopted as the prior for the $B^{e}$ matrix. The same holds the $\Gamma^{e}$ matrix.

The posterior inference of the distribution of Equation (Equation 8) cannot be computed analytically, but an approximation can be found. In this work we use Automatic Differentiation Variational Inference (ADVI) [104] to compute the approximate posterior distribution.

The inference output for each event in each scan path performed by a given subject are the posterior PDFs over the covariance matrices $B^{e}$ and $\Gamma^{e}$. Such distributions are summarised by computing their sample average and Highest Density Interval (HDI). The distribution summaries related to the diagonal and one of the off-diagonal components of the matrices (recall that B and Γ are symmetric matrices) are joined together, thus yielding the vector $v_{\left(id\right)}^{e}$ for each subject $id\in \left[1,\ldots ,ID\right]$, ID being the total number of subjects:(9)$$v_{\left(id\right)}^{e}=\left[B_{ii}^{avg,e},B_{ii}^{hdi,e},B_{ij}^{avg,e},B_{ij}^{hdi,e},B_{jj}^{avg,e},B_{jj}^{hdi,e},\Gamma_{ii}^{avg,e},\Gamma_{ii}^{hdi,e},\Gamma_{ij}^{avg,e},\Gamma_{ij}^{hdi,e},\Gamma_{jj}^{avg,e},\Gamma_{jj}^{hdi,e}\right].$$

In simple terms, any fixation or saccade is compactly described through the inferred means of parameters $B_{ii},B_{ij},B_{jj},\Gamma_{ii},\Gamma_{ij},\Gamma_{jj}$ and related uncertainties ($B_{ii}^{hdi,e}$, etc.; but see Table 2, for a summary). Such information is then collected for each event composing the scan path.

In brief, we eventually have at hand the characterisation of the visual behavior of observer id while scrutinising the stimulus *k* (image), captured as a sequence of events (the fixations and saccades composing the scan path), each event *e* being summarised by the vector $v_{(id),k}^{e}$.

### 5.3. Identity Recognition from Gaze Dynamics

The identity recognition procedure is depicted at a glance in Figure 4. After the event classification and O-U SDE parameters inference described above, two different models for fixational and saccadic movements are trained separately from the inferred O-U parameters to discriminate between identities. The predictions from the two models are then fused at the score level, thus achieving the final classification.

More specifically, in the present work a Support Vector Machine (SVM) is employed for both the fixation and saccade models. SVMs are trained to discriminate between the ID identities from the $v_{\left(id\right)}^{e}$ vectors describing the dynamics of each event.

For a One vs. Rest SVM the id-th decision function on the event *e* is:(10)$$Df_{\left(id\right)}^{e}=w_{\left(id\right)}^{T}\phi \left(v^{e}\right)+b_{\left(id\right)},$$
where $w_{\left(id\right)}$ is parameter vector for the id-th classifier ($b_{\left(id\right)}$ being the bias) and ϕ(.) is the kernel function. Let $Dv^{e}=\left[Df_{\left(1\right)}^{e},\ldots ,Df_{\left(id\right)}^{e},\ldots ,Df_{\left(ID\right)}^{e}\right]$ be the decision vector collecting the decision function values at each class for the event *e*. Define
(11)$$\langle Dv^{fix}\rangle =\frac{1}{F}\sum_{a=1}^{F}Dv^{e_{a}^{fix}};\;\langle Dv^{sac}\rangle =\frac{1}{S}\sum_{a=1}^{S}Dv^{e_{a}^{sac}}$$
the average decision vectors for all the fixations and saccades composing one trajectory. The final classification of the identity Id associated to the trajectory is obtained as (score fusion):(12)$$Id=argmax_{id}\left\{\langle Dv^{fix}\rangle +\langle Dv^{sac}\rangle \right\}$$

## 6. Experimental Results and Analyses

Here we provide a quantitative evaluation of the method in order to address the research question pinpointed in Section 1: is such a method based on a minimal, albeit general, phenomenological model of gaze behaviour adequate for biometrics?

First, a straightforward standard evaluation in terms of identification performance is provided.Second, an analysis is carried out, which aims at gaining a deeper insight into the features made available by the method and the identification results so far achieved.

The experimental assessment of the method exploits the Fixations in Faces (FIFA) eye-tracking dataset [15] that is briefly described below.

### 6.1. Dataset

The FIFA dataset consists of eight subjects whose eye movements are recorded during the free-viewing of 200 1024 × 768px images for 2 s each. Eye movements are recorded by an Eyelink 1000 eye-tracker with a sampling frequency of 1000 Hz.

In spite of the modest size of the subjects’ sample, this dataset is appealing for our purposes as regards the type of stimuli, the task, and the duration of the recording session. As previously discussed, under a free-viewing task, static images are susceptible to elicit idiosyncratic behaviour from the observer [68]. In addition, the stimulus presents elements of semantic complexity suitable to trigger high-level cognitive control of visual attention [15]. Images include salient objects and many different types of faces. This data set was originally intended to quantify the attractiveness of human faces for observers and to test salience models including face detectors.

Two-second recordings at 1000 Hz sampling frequency represents a viable trade-off between collecting a gaze trajectory of informative length and biometrics time constraints [36,43]

### 6.2. Identification Performance

The proposed approach has been evaluated via a 10-fold cross validation procedure; Table 3 reports the results in terms of 4 evaluation metrics, namely Accuracy, F1 Score, Equal Error Rate (EER) and Area Under the Curve (AUC). For comparison, the same procedure has been adopted for the algorithms proposed in [106,107].

In [106], George and Routray extract a large number statistical features (12 features for each fixation and 46 features for each saccade) from the position, velocity and acceleration profiles of the gaze sequence plus the computation of features related to the duration, dispersion, path length and co-occurrence of fixations and saccades. Such high-dimensional feature set is then necessarily reduced via backward feature selection, which retains 9 and 43 features for fixations and saccades, respectively. The obtained feature set is then fed into a Gaussian Radial Basis Function Network (GRBFN).

Schröder et al. [107] improved on the results obtained by George and Routray [106] using their original feature set and a Random Decision Forest as the classification algorithm. Notably, Schröder et al. [107] released the implementation code for their approach and the one proposed in [106].

Table 3 shows the 10-fold cross-validated results of the proposed method and those adopted by [106,107]. Notably the proposed approach exhibits performances comparable to the others, despite the adoption of a more succint feature set. Indeed, the composite O-U model is fully specified by the parameters $B^{e}$ and $\Gamma^{e}$ (with e∈[fix,sac]): no feature selection stage is needed and each parameter has a clear interpretation (cfr. Table 2). The posterior distribution summaries (mean and HDI) of these matrices, collected into the $v^{e}$ vector, yield a feature set whose dimensionality, if compared with [106,107], is more than halved.

Figure 5 depicts the micro-averaged ROC Curve on the validation sets of one of the 10 folds. AUC and EER are reported on the bottom-right. The confusion matrix on the 8 available identities is shown in Figure 6 (left), while the Cumulative Match Score (CMS) curve is displayed on the right. The latter demonstrates the ranking capabilities of the proposed methodology, as the correct identity is found in the first 3 ranked predictions with remarkable accuracy (Demo code and data is available on the website: https://github.com/phuselab/Gaze_4_behavioural_biometrics, accessed on 14 January 2023).

For what concerns the computational cost of the whole process, it is clearly dominated by the Bayesian inference procedure adopted for the estimation of the O-U SDE parameters associated to each event (crf. Equation (Equation 8)). The computational cost of scan path event classification via SVM can be considered negligible. In our experiments processing a single event (i.e., approximate posterior inference via ADVI) took on average ∼9 seconds adopting the following hardware configuration: Intel Core i7-8700K 4.70 GHz CPU, 16GB of RAM, while the software implementing approximate inference relies on the pyMC3 library [108]. The cost of processing an entire scan path grows linearly with the number of events composing it. Significant speed-up improvements could be eventually obtained by employing dedicated hardware (e.g., GPUs) and optimised software implementation, e.g., adopting JIT compilation and state of the art computational back-ends (e.g., JAX). Yet, a broad inquiry about the hardware/software optimisation choices for speed-up improvement is out of the scope of this work.

### 6.3. Analysis of Results

A classic bare evaluation of the identification performance does not fully make justice of the role played by the parameters/features in the end-to-end scheme outlined in Figure 4. This section presents some questions weighing the model-based parameters in the service of biometrics and assessments aimed at addressing them.

To this end, recall that for each subject id, and for each image *k* in the dataset, we have a corresponding scan path defined as the collection of events *e* (fixation, saccade), each event being described by the vector $v_{(id),k}^{e}$ as defined in Equation (Equation 9). For clarity’s sake, the physical meaning of vector components is summarised at a glance in Table 2.

Denote:$\langle v_{(id),k}^{fix}\rangle$ the average fixation feature vector and $\langle v_{(id),k}^{sac}\rangle$ the average saccade feature vector associated to the scan path *k*:
(13)$$\langle v_{(id),k}^{fix}\rangle =\frac{1}{F}\sum_{a=1}^{F}v_{(id),k}^{e_{a}^{fix}},\;\langle v_{(id),k}^{sac}\rangle =\frac{1}{S}\sum_{a=1}^{S}v_{(id),k}^{e_{a}^{sac}};$$$v_{(id),k}$ the descriptor of scan path *k* obtained by concatenating the two vectors above:
(14)$$v_{(id),k}=\left[\langle v_{(id),k}^{fix}\rangle |\langle v_{(id),k}^{sac}\rangle \right];$$$\langle v_{\left(id\right)}\rangle$ the summary descriptor of the visual behaviour of observer id, over the set of the *K* observed stimuli:
(15)$$\langle v_{\left(id\right)}\rangle =\frac{1}{K}\sum_{k=1}^{K}v_{(id),k}.$$

*How do the model-based descriptors differentiate subjects?* A preliminary description of how the proposed model captures the attentive behaviour of the individual observer id is provided by the summary vector $\langle v_{\left(id\right)}\rangle$ defined in Equation (Equation 15). Qualitatively, this can be compared against the vector describing the other observers through the heatmap presented in Figure 7.

The rows of the heatmap show how the vector components/features vary between subjects. It can be easily appreciated how each row captures a behavioural “thread”, a signature that is unique to each observer (this straightforward view can be further analysed to some extent, e.g., through the violin plots of each feature per subject as in the Appendix A).

What is the degree of differentiation between individuals? Beyond visual evaluation, one may question whether such individual signatures do or do not correlate with one another. This information is summarised in Figure 8 through a coloured Hinton diagram (named after Geoff Hinton, who used this type of display to plot the weight matrix of a neural network).

In the diagram, the size of each square represents the magnitude of the correlation between two subjects, say id=i and id=j,i≠j, in terms of the correlation $\rho (\langle v_{\left(i\right)}\rangle ,\langle v_{\left(j\right)}\rangle )$, while the colour renders the correlation/anticorrelation spectrum. Overall, this result shows that, for the data we are dealing with, there is a low correlation among individuals’ visual behaviour when gauged via the proposed model-based descriptors.

Are the individual descriptors stable with respect to different stimuli? The degree of feature correlation that can be measured between different observers is one aspect. The other, which is deemed to be important for biometric purposes, is the observer’s parameter consistency or low behavioural variability over the set of viewed stimuli.

This second aspect (within- or intra-subject variability) can be evaluated in terms of the distributions of correlation coefficients calculated for subject *i* over different stimuli *k* and *l* of the stimulus set; namely, $\rho_{intra}\left(\langle v_{(i),k}\rangle ,\langle v_{(i),l}\rangle \right)$, k,l=1⋯K,k≠l.

Between- or inter-subject variability is similarly characterised by considering the distribution of correlation coefficients $\rho_{inter}\left(\langle v_{(i),k}\rangle ,\langle v_{(j),l}\rangle \right)$, k,l=1⋯K,k≠l,i≠j,

For either ensemble of correlation coefficients an empirical estimate of the ensemble distribution can be obtained via Kernel Density Estimation (KDE), say $\tilde{P}\left(\rho_{intra}\right)$ and $\tilde{P}\left(\rho_{inter}\right)$. The outcome of this analysis is presented in Figure 9.

By and large, it is shown how inter-subject correlation is described by a wide, Gaussian-like distribution (denoting high variability) as opposed to intra-subject correlation that is captured by a remarkably peaked distribution (low variability).

What is learnt from individuals’ parameters given a set of stimuli? For a group of subjects and a finite set of stimuli eliciting viewing behaviours, an overall representation of such behaviours can be attained by considering each vector $v_{(id),k}$ a point on a high dimensional manifold. To gain understanding of such manifold, what we need first is to construct a topological representation of the high dimensional data. Subsequently, we need to project such manifold on a lower dimensional one for providing a suitable visualization of the manifold’s patterns and structure.

Uniform Manifold Approximation and Projection (UMAP) is one suitable tool for this purpose. Based on Riemannian geometry and algebraic topology [109,110], it aims to preserve local and global structure in the data; in this respect UMAP provides better results than other sophisticated dimensionality reduction techniques such as t-SNE [111]. The preservation property is helpful when exploring relationships within the data that may otherwise be lost in other data transformations. To such end, UMAP first constructs a graph-based topological representation of the high-dimensional data and uses it to optimise its equivalent low-dimensional topological representation [110]. This implies a low-dimensional data projection that is highly correlated to the original high-dimensional representation. Within this optimisation, key hyperparameters are selected that alter the weighting of global or local data preservation. For this study, we first adopted the semi-supervised version of UMAP to perform dimensionality reduction in order to visualize the subjects’ behaviours (points $v_{(id),k}$) in the high dimensional feature space. UMAP parameters were chosen as *n* neighbours=20, min dist=0.99, and the lower order dimension was set to 2. The semi-supervised approach exploited only a fraction (50%) of the available labels (subject ids). The procedure has been repeated by learning the manifold for a varying dimensionality of the feature vector $v_{(id),k}$. In three trials, we considered the 20%, the 50% and the full set of vector components; in the first two cases, components have been selected by sampling from a uniform distribution. Results presenting the projected 2-D manifolds are depicted in Figure 10.

As expected, the adoption of a subset of features yields more uncertain (Figure 10b) or even indistinguishable borders (Figure 10a) circumscribing subject’s representations in the manifold. This result relates to the completeness of the proposed model-based feature set.

In spite of the fact that in the present work the choice of the classifier is instrumental in providing a testable system in terms of standard benchmark metrics, it is interesting to exploit the manifold representation for visualising the effect of the classification step.

This can be achieved by learning the manifold of the classified subjects’ behaviours. In this case, a point in the manifold is represented by the decision vector over a stimulus *k* obtained at the Score Fusion stage on the test set (cfr. Equation (Equation 12)).

Since classification results are obtained going through the train/test stages, the outcome vectors of the testing stage have been directly used for learning the manifold via the unsupervised version of the UMAP. The result is presented in Figure 11b; it can be compared with the original manifold in Figure 11a (already shown as Figure 10c).

The latter result provides a tangible account for both the classifier effect on the topological structure of the manifold and its suitability for learning subjects’ separation hypersurfaces within the classifier induced manifold.

## 7. Discussion and Conclusions

In this paper we laid down a principled approach to the analysis of eye movements for behavioural biometrics and derived, in the context of such framework, a computational model suitable to operationalise biometric assessment. The approach grounds in foraging theory, which provides a sound basis to capture the uniqueness of individual eye movement behaviour; further, it is appropriate to pave the way for deriving models of gaze behaviour analysis. In particular, we proposed a composite Ornstein-Uhlenbeck process for quantifying, at the phenomenological level, the exploration/exploitation signature characterising the foraging eye behaviour. The relevant parameters of the composite O-U model, inferred from eye-tracking data via Bayesian analysis, are shown to yield a suitable feature set for biometric identification; the latter here, for demonstration purposes, is eventually accomplished via a classical classification technique (SVM).

To the best of our knowledge, the work presented here is novel in aiming at a sound, model-based approach to the exploitation of eye movements for biometric purposes. We argue that understanding the principles that underlie the deployment of gaze in space and time is important for offering this field a fresh view on either the analyses of data and prospective applications.

### 7.1. Advantages of the Method

The merit of the method stems from straightforwardly confronting the very question of visual attention deployment through gaze: *Where to look next?*. Crucially, the “where” part concerns the selection of *what* to gaze at—features, objects, actions—and their location within the scene; the “next” part involves *how* we gaze at what we have chosen to gaze, a fundamental issue by and large neglected in the literature concerning the computational modelling of attention (but for a critical discussion, see [68,95,112,113,114]).

It is no surprise that exactly the *how* problem was posed since the earliest proposal by Kasprowski and Ober [36] that considered eye movements for biometrics. In order to retain information on how the individual looks, while factoring out details concerning where the person is looking, they exploit a simple “jumping point” kind of stimulation. Clearly, such choice has the drawback to rule out, as the authors lucidly pointed up, important individual information, somehow collapsing brain activity behind gaze to the crude oculomotor system activity. On the other hand, it has the advantage of avoiding, to some extent, the learning/habituation effect (the brain learns the stimulation and acts differently after several repetitions of the same visual stimulus [36]).

Here, in a different vein, rather than simplifying the stimulus, we put effort on a principled modelling of the *how* dynamics. Hence, the presented results have been achieved by considering free-viewing observation of a number of different static images per subject, depicting complex scenes embedding a variety of “semantically rich” objects/regions.

### 7.2. Limitations

The overall quantitative outcome of our proof of concept encourages future effort in this direction. Yet, the present study is not innocent of limitations, first of all, the number of subjects involved in the dataset we employed. Motivation for this choice was discussed in Section 6.

Provided that not only classified event data (fixations and saccades) but raw eye-tracking data are available, a number of publicly available datasets could be adopted to the purpose of widening the sample size, e.g., DOVES [115], EMOd [116] and, recently, GazeBase [117], EyeT4Empathy [118]. Such datasets collect a variety of data as regards the recording duration, kinds of stimuli and given task. For example, GazeBase includes data recorded through a battery of seven tasks and related stimuli (fixation task, horizontal saccade task, oblique saccade task, reading task, free viewing of cinematic video task, and gaze-driven gaming task); EyeT4Empathy considers free exploration of structureless images and gaze typing (writing sentences using eye-gaze movements on a card board). Obviously, given a dataset, one could barely choose to benchmark the method under investigation against a specific task/stimulus type. However, more appealing would be the comparison across tasks, either to assess the generalisability of the method or to select the optimal stimulus in view of a practical application. What are the precise metrics for such conditions is far from obvious. Further, one might consider a cross- or multiple-dataset evaluation, a cogent methodological problem that has been recently raised in machine learning and computer vision [119,120,121,122,123] together with the growing quest for appropriate statistical procedures in performance assessment. In the field of gaze-based biometrics, this issue turns into an open research question *per se*.

Beyond benchmarking aspects, it is clear that there is yet an engineering gap between the modelling effort and the practice of eye movement analyses in real world biometric applications. Performance is not the only criterion that is to be taken into account. Biometric systems should be easy to use, have low cost, be easy to embed and integrate in the target security application and be robust and secure [124]. This, in turn, involves again the factors we have mentioned (stimuli, tasks, etc.), and in what follows they are discussed and to some extent related to the work presented here.

### 7.3. Session Duration

To be useful in biometric identification, a single experiment/session should be as short as possible [36,43]; more precisely, it has been recommended that check time duration should be less than 12 secs [43]. Here, we used 2 secs recording sequences; however, from a general standpoint, further work is needed to assess the dependency of check time on kinds of stimuli and task requested to the user. As a matter of fact, the kind of stimulus and task to be adopted are still open questions in this field.

### 7.4. The Stimulus Problem

We have previously touched on some implications of the nature of the stimulus that concern the nuances of the information eventually inferred. Eye movements, in their *what* component, are strongly correlated with the kind of scene the subject is looking at. More sophisticated stimulation could be in the form of a dynamic one (e.g, a short clip or animation). The method presented here, both in the initial parsing of gaze raw data via the NSLR-HMM algorithm [100], and in the subsequent O-U analysis is capable of handling, in its present form, dynamic sequences (that also bring forth smooth pursuit gaze shifts). In the perspective laid down in this work, this choice would provide further information to unveil the distinctive, subjective nature of gaze deployment. But again, it should depend in turn on the choice of the specific biometric application. Interestingly enough, even data collected through a wearable eye-tracker could in principle be employed in specific contexts; an opportunity—beyond the mere evaluation of fixation points collected while the user is scrutinising a screen-, which has not yet been considered in this field.

### 7.5. The Task Problem

It is well known that gaze is not generically deployed to objects but allocated to task-relevant objects within the scene; thus deployment may or may not correlate with the salience of regions of the visual array [1,125,126,127,128]. In this respect, recent theoretical perspectives on active/attentive sensing [14,129] have highlighted the fundamental role of the exploitation/exploration pattern, which is a fundamental tenet grounding the research presented here. The analyses we have discussed rely on recordings under a free-viewing condition. Free-viewing is generally defined as an experimental paradigm where the viewer is given no specific instructions during the experiment other than to look at the images. Yet, the approach is deemed to be controversial in classical eye-movement studies [68]. It has been argued that free-viewing simply gives the subject free license to select his or her own internal task/goal/priorities [68]. As a consequence, viewing behaviour is not studied while free of task, but rather while having no real knowledge of the subjective purposes of looking. But interestingly, the individual’s idiosyncrasy that arises from this experimental choice, which is problematic in classical visual attention studies, paves the way, for biometric aims, to capture the individual’s uniqueness, as we have discussed from the beginning. Indeed, defining what is a goal is far from evident. The classic visual attention dichotomy between top-down and bottom-up control assumes the former as being determined by the current “endogenous” goals of the observer and the latter as being constrained by the physical, “exogenous” characteristics of the stimuli (e.g.,flashes of light, loud noises, independent of the internal state of the observer). The construct of “endogenous” attentional control is subtle since it conflates control signals that are ”external” (induced by the given current task voluntarily pursued) “internal” (such as the motivation for paying attention to socially rewarding objects/events), and selection history (prioritising items previously attended in a given context). In terms of the exploitation/exploration pattern, which is generally related to maximize expected rewards, or, equivalently, to minimize the future expected surprise [14,129], a distinction should be acknowledged between “external” rewards (incentive motivation, e.g, monetary reward) and reward related to “internal” value. The latter has different psychological facets [130] including affect (implicit “liking” and conscious pleasure) and motivation (implicit incentive salience, “wanting”). Eliciting these aspects can be strategic for biometric purposes.

On the other hand, the choice of an external task/goal might offer novel avenues for biometrics. In point of fact, the choice needs not to be reduced to viewing instructions provided to the user. It has been suggested that, for real-life applications, in the future, eye-movement biometrics should be applied jointly with some other biometrics, such as face or pupil images or fingerprints [131]. Further, rather than limiting to some kind of multi-modal fusion, the task might consider, in an appropriate application context, the specification of an actual joint activity involving both gaze and other gestures. For instance, eye-hand coordination, e.g., in a drawing/copying task, could provide a suitable visuo-motor signature to characterise the subject [132,133].

### 7.6. Handling Collected Eye-Tracking Data

A comment is deserved to the parsing step used to transform raw gaze trajectories into a discrete sequence of events, namely fixations and saccades (for static images; smooth pursuit should also be considered for image sequences). Albeit eye movements have been measured since the early 1900s, the event classification step conceals subtleties either at the conceptual level (how to define fixations and saccades) and at the algorithmic level that operationalises the previous (but for an in-depth discussion, the reader is urged to refer to Hessels et al. [99]). Unfortunately, this issue is by and large overlooked in the biometric literature with few exceptions, e.g., [134] who devoted effort to analise the effect on final identification due to the widely adopted, baseline parameters, namely velocity threshold and minimum fixation duration. This attitude is somehow surprising, given the circumstances, as noted in Section 3, that the vast majority of methods barely rely on saccade/fixation data. As to the work presented here, on the one hand we adopt the NSLR-HMM algorithm [100] that differs in concept from the traditional workflow of pre-filtering, event detection and segmentation; it allows for experiments with complex gaze behaviour and can be used on both high-quality lab data and more challenging mobile data on natural gaze behaviour, while requiring minimal parameter setting.

On the other hand, in our approach, event classification is instrumental for a coarse-grained segmentation of gaze dynamics into exploration/exploitation phases, whilst parameter inference is primarily based on raw trajectories. Overall, the approach mitigates the dependency of the final identification results on the initial event parsing algorithm and its constitutive, most often empirically chosen thresholds/parameters.

A conclusive note concerns the generality of the method and the derived parameters/features. The fact that the method yields a “behavioural signature” in terms of a set of explainable parameters that compactly recapitulate the individual’s gaze dynamics, paves the way for potential applications in research fields where one such signature is needed. A clear example is psychiatry [135] where eye movement analysis can potentially be used and applied to help refining or confirming a psychiatric clinical diagnosis. In this realm, internal consistency and temporal stability of gaze events (e.g., saccades) have gained currency to define endophenotypes of specific disorders [136]; furthermore, a signature in terms of explainable parameters is likely to help interpreting distorted eye movements in psychiatric patients in terms of functional brain systems. Other examples can be easily gathered from recent developments in emotion [137,138,139] and personality research [72,140,141,142,143].

## Figures and Tables

**Figure 2 sensors-23-01262-f002:**
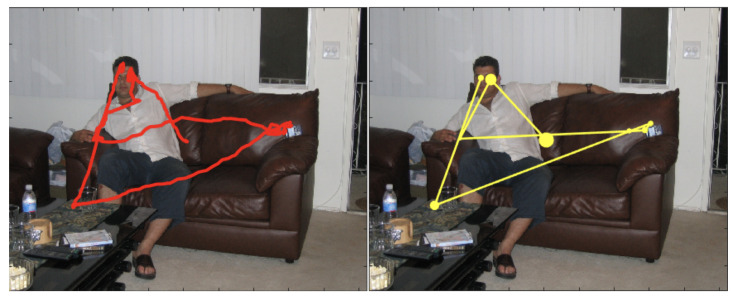
The dynamics of one observer’s gaze while visually foraging on the image landscape as recorded through an eye-tracking device. **Left**: the “raw data” represented as a time-sequence of spatial coordinates (red dots) displayed as the red trace overlapped on the viewed image. **Right**: the observer’s scan path, namely, the continuous raw data trace parsed into a discrete sequence of fixations (yellow disks) and saccades (segments between subsequent fixations); disk radius is proportional to fixation time. Image and eye-tracking data are publicly available from the FIFA dataset [15].

**Figure 3 sensors-23-01262-f003:**
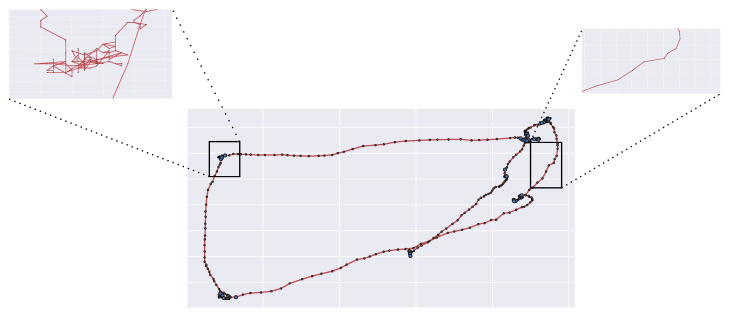
A zoom-in on fixations and saccades as recorded by high frequency eye-trackers.

**Figure 4 sensors-23-01262-f004:**
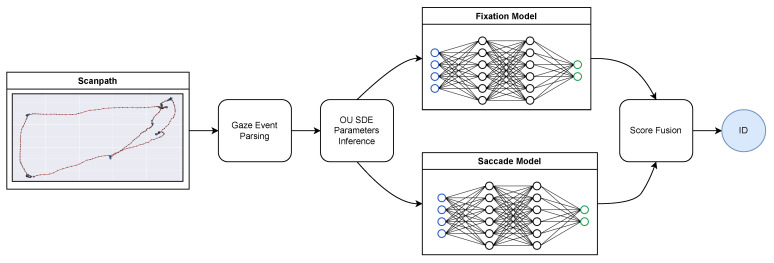
The proposed approach at a glance.

**Figure 5 sensors-23-01262-f005:**
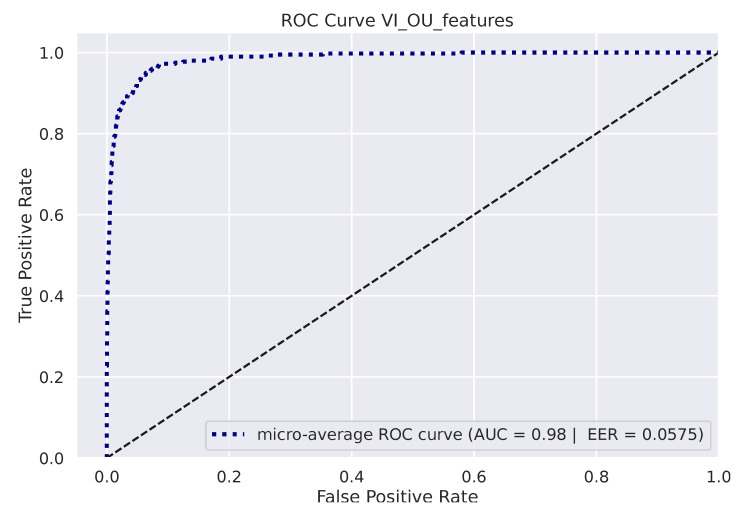
Receiver Operating Characteristic (ROC) curve.

**Figure 6 sensors-23-01262-f006:**
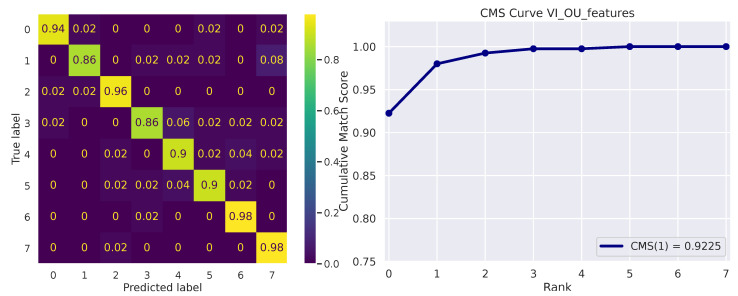
Confusion matrix (**left**) and Cumulative Match Score (CMS) curve (**right**).

**Figure 7 sensors-23-01262-f007:**
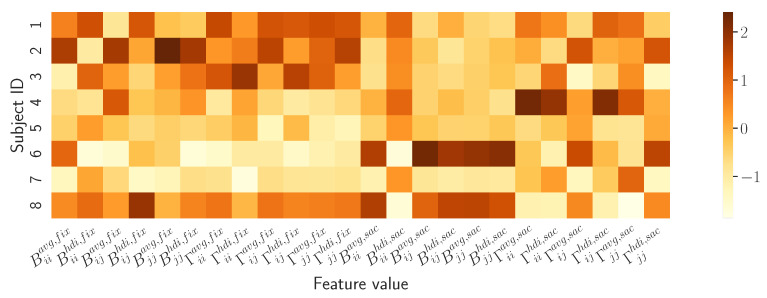
Heatmap contrasting the model-based descriptors of observers’ visual behaviour. Each row of the map renders the summary vector $\langle v_{\left(id\right)}\rangle$ for subject id; numeric values of vector components are associated to colours as depicted in the side bar.

**Figure 8 sensors-23-01262-f008:**
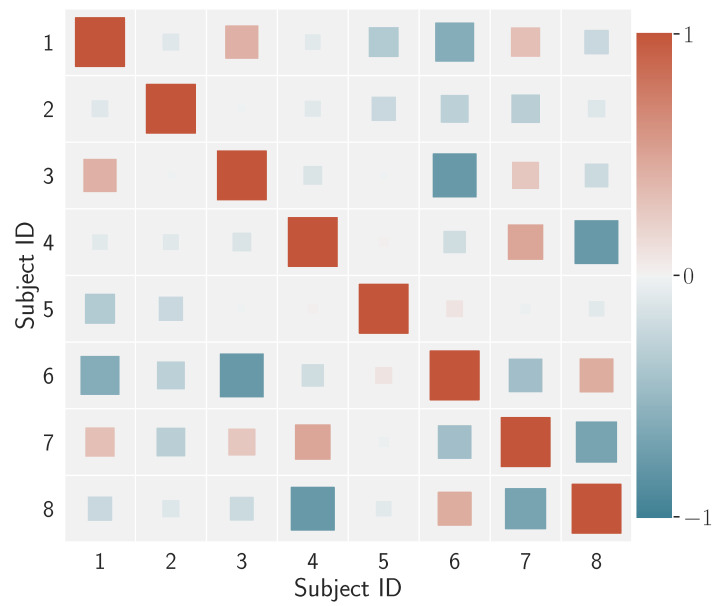
Hinton diagram for the correlation coefficients calculated over the average feature vectors of the subjects. The square size is proportional to the magnitude of the correlation; the colour denotes the degree of the correlation coefficient in the range depicted in the side bar.

**Figure 9 sensors-23-01262-f009:**
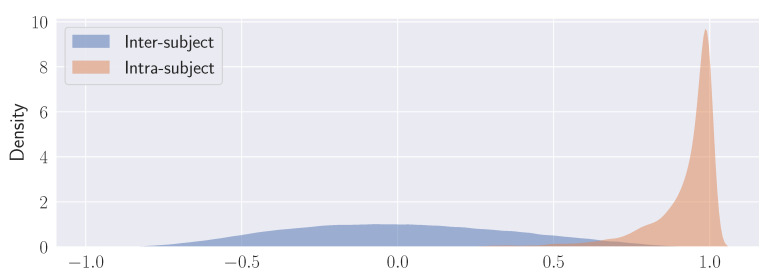
Kernel density estimation (KDE) plots for the distributions of the correlation coefficients calculated over all the stimuli for each subject (Intra-subject) and between subjects (Inter-subject). See text for explanation.

**Figure 10 sensors-23-01262-f010:**
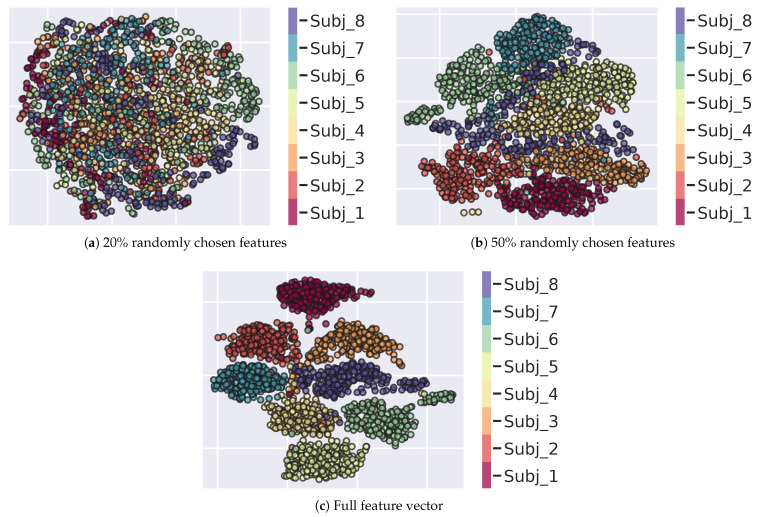
(**a**) Semi-supervised UMAP 2D projection of a subset of 20% of the original O-U parameters. (**b)** Semi-supervised UMAP 2D projection of a subset of 50% of the original O-U parameters. (**c**) Projection obtained by using the full feature vector. In the (**a**,**b**) cases, the subset of features is constructed by selecting features at random via uniform distribution sampling. Each point represents a subject (identified by a colour in the side bar) observing a specific image.

**Figure 11 sensors-23-01262-f011:**
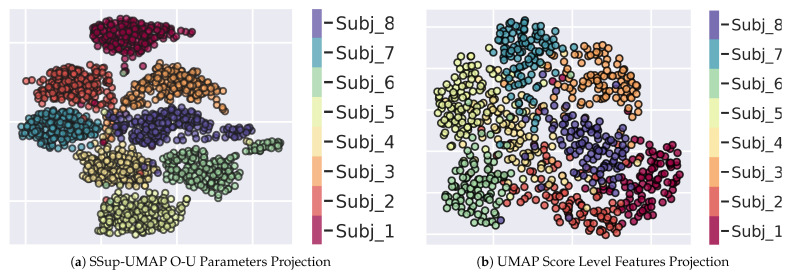
(**a**) Semi-supervised UMAP 2D projection of the concatenation of Fixations and Saccades O-U parameters. Each point represents the parameters extracted from one subject on an image of the FIFA dataset; the different point of the same color indicate the same subject observing different images. (**b**) UMAP 2D projection of vectors from the Score level fusion representation. Each point represents a subject (identified by colour in the side bar) observing a specific image.

**Table 1 sensors-23-01262-t001:** Relationship between Visual Attention and Foraging.

Visual Attentive Processing of a Scene	Patchy Landscape Foraging
Perceiver	Forager
Perceiver’s gaze shift	Forager’s relocation
RoI/object	Patch
RoI/object selection	Patch choice
Deploying attention to RoI/object items	Patch/prey handling
Disengaging from RoI/object	Patch leave or giving-up

**Table 2 sensors-23-01262-t002:** Summary of the proposed features and related physical interpretation.

Dimension	Description
$B_{ii}^{avg,e}$ ($B_{ii}^{hdi,e}$)	The element in position ii of the posterior distribution mean of the inferred $B^{e}$ matrix. It represents the drift (strength of the attraction) towards the attractor point (arriving/current fixation) in the horizontal direction. High $B_{ii}^{avg,e}$ values denote higher velocity saccades or less pronounced physiological drift (see [105]) in fixational eye movements (FEMs) in the *i* dimension. ($B_{ii}^{hdi,e}$: uncertainty of the estimation)
$B_{ij}^{avg,e}$ ($B_{ij}^{hdi,e}$)	The element in position ij of the posterior distribution mean of the inferred $B^{e}$ matrix. It represents the cross-correlation between the drifts in the horizontal and vertical dimension. High $B_{ij}^{avg,e}$ values denote more curved saccades/FEMs. ($B_{ij}^{hdi,e}$: uncertainty of the estimation)
$B_{jj}^{avg,e}$ ($B_{jj}^{hdi,e}$)	The element in position jj of the posterior distribution mean of the inferred $B^{e}$ matrix. It represents the drift (strength of the attraction) towards the attractor point (arriving/current fixation) in the vertical direction. High $B_{jj}^{avg,e}$ values denote higher velocity saccades or less pronounced physiological drift (see [105]) in FEMs in the *j* dimension. ($B_{jj}^{hdi,e}$: Uncertainty of the estimation)
$\Gamma_{ii}^{avg,e}$ ($\Gamma_{ii}^{hdi,e}$)	The element in position ii of the posterior distribution mean of the inferred $\Gamma^{e}$ matrix. It represents the amount of diffusion exhibithed by a fixation/saccade in the horizontal direction. High $\Gamma_{ii}^{avg,e}$ values denote higher noise in saccades/FEMs in the *i* dimension. ($\Gamma_{ii}^{hdi,e}$: uncertainty of the estimation)
$\Gamma_{ij}^{avg,e}$ ($\Gamma_{ij}^{hdi,e}$)	The element in position ij of the posterior distribution mean of the inferred $\Gamma^{e}$ matrix. It represents the covariance of the diffusion between horizontal and vertical dimension. High $\Gamma_{ij}^{avg,e}$ values denote more anisotropic saccades/FEMs. ($\Gamma_{ij}^{hdi,e}$: uncertainty of the estimation)
$\Gamma_{jj}^{avg,e}$ ($\Gamma_{jj}^{hdi,e}$)	The element in position jj of the posterior distribution mean of the inferred $\Gamma^{e}$ matrix. It represents the amount diffusion exhibited by a fixation/saccade in the vertical direction. High $\Gamma_{jj}^{avg,e}$ values denote more noise in saccades/FEMs in the *j* dimension. ($\Gamma_{jj}^{hdi,e}$: uncertainty of the estimation)

**Table 3 sensors-23-01262-t003:** The obtained results in terms of the Accuracy, F1 Score, Equal Error Rate (EER) and Area Under the Curve (AUC) metrics.

	Accuracy	F1 Score	EER	AUC
Proposed	93.75%±3.4	93.68%±3.5	0.046±0.01	98.98±0.4
[106]	92.06%±3.2	91.95%±3.3	0.048±0.01	98.88±0.4
[107]	92.93%±2.4	92.73%±2.5	0.039±0.01	99.25±0.2

## Data Availability

Data and code for reproducing the experiments reported in this article are available at this link https://github.com/phuselab/Gaze_4_behavioural_biometrics, accessed on 14 January 2023.

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
