# Peer review of "Using Gaze for Behavioural Biometrics"

_sensors, 2023, doi:10.3390/s23031262_

Round 1
Reviewer 1 Report
I enjoyed reading this paper. I use image recognition in a different context, but I am not an image recognition specialist. Nonetheless, I found the article interesting. It is well-written and well-organized. It looks into the case from a different angle. However, I guess it will interest only a particular, perhaps small, community, but maybe that is what the authors are also expecting.
Although I can address some typos (from my perspective) or some changes in the presentation and organization of the manuscript, I prefer not to destroy the joy of accepting a paper as it is for the first time.
Well done, colleagues!
Author Response
We thank the Reviewer for the general consideration of our work.
Reviewer 2 Report
Review: Using Gaze for Behavioral Biometrics
General:
As is, this paper is unreadable and without any discernable focus. If it can be brought into a publishable form, it needs to be shortened significantly and focused on the question at hand. At times it reads like philosophical treatise, a rediscovered Freudian text, or worse…. (e.g., “uniqueness of the self”, “capture the ineffable something”).
Although Marcel Proust was a great writer, such references have no place in a scientific paper.
The first 11 pages would need to be condensed into maybe 3-4 pages and focused on the salient points (up to section 6). Then we have one (1) page of Results (+ figures), and again 2.5 pages of unreadable “Discussion”.
Language: it seems that translation soft-ware was used that chose words that are not commonly used in papers of the theme at hand.
Specific points:
Abstract: The first sentence should be in passive mode.
l.10: “..and on prospective applications….”
l. 14: You cannot start a paper with “Consider Figure 1”
l. 14 What is “wandering gaze”
l. 23: What is a “complex individual”
l.90: “latterly” ???
l. 183: “more neatly” ???
l. 506: “in a different vein” ???
Reviewer 3 Report
The manuscript is clean and clear. However, the authors should add the research questions that they are solving along with the contribution of the research work.
Author Response
We are grateful to the Reviewer for the general appreciation of our effort and constructive suggestions. The research question has been made clear in the introduction and referred to in the novel part of the experimental section.
Reviewer 4 Report
To authors:
The authors reported an interesting study about behavior biometrics for gaze analysis. The first part of the manuscript reviewed extensively on this topic and reasoned clearly about the background and motivation for their proposed approach to evaluate the Bayesian inferred gaze parameters from the dynamic eye movement model using the composite Ornstein-Uhlenbeck process. The provided results evaluating the author's approach were compatible with prior methods. The manuscript is well-written and a great read.
First of all, I would like to congratulate the authors on their effort in this excellent study. However, a few minor issues regarding the manuscript will be listed below.
1. There are a few typos, misplaced punctuations, and undesignated abbreviations/formulas, in lines 252, 296, 377, 410, 416, 418, 434, 535, 536, and 582. Please recheck the whole manuscript for these minor errors.
2. As stated above, the formulas after formula 6 were numbered wrongly due to an undesignated formula between lines 369 and 370. Please adjust accordingly.
3. The authors laid out in length the rationale behind the proposed algorithm, but the results given are considered scarce (only 8 subjects in the Fixations in Faces (FIFA) dataset) compared to the whole narrative. The authors should elaborate more on their results and maybe include other experiments on larger datasets for assessment, such as GazeBase, EyeT4Empathy, DOVES, OSIE, VIU, EMOd…etc. It would be interesting to know the performance of the proposed method in these diverse datasets with different eye-tracking devices.
4. In the Result section [6. Experimental work and results], the authors compared their proposed approach with two other prior published methods. In lines 465-467, the authors claimed that their feature sets were more succinct than the prior methods. Please provide the numbers of the feature sets and the computer rendering time between these methods to add more volume and practical implications to the authors’ claim.
5. While reading into this topic, a thought had come up. At the Score Fusion stage of the proposed approach (Figure 4), since the author proposed a phenomenological method to quantify the exploration/exploitation of foraging eye behavior, would it be possible to not just classify the characteristic of the individual with the feature sets but to somehow visualize (heatmap or feature-map) the parameters to represent the individual’s characteristics or event status of health (with an additional parameter such as general condition, attentiveness, tiredness.. taken into account)? I would be very interested to know more about the development of this approach in the field of behavior biometrics and all future psycho-sociological experiments or medical implications this approach can bring us.
Round 2
Reviewer 2 Report
The paper significantly improved. It seems the authors have incorporated all suggestions and criticisms.